# h-BN Modification Using Several Hydroxylation and Grafting Methods and Their Incorporation into a PMMA/PA6 Polymer Blend

**DOI:** 10.3390/nano12162735

**Published:** 2022-08-09

**Authors:** Abdelwahab Boukheit, France Chabert, Belkacem Otazaghine, Aurélie Taguet

**Affiliations:** 1Polymers Composites and Hybrids (PCH), IMT Mines Ales, 30319 Ales, France; 2Laboratoire Génie de Production (LGP), ENIT-INPT University of Toulouse, 65000 Tarbes, France

**Keywords:** polymer nanocomposites, boron nitride, surface modification, silane grafting, co-continuous morphology, selective localization

## Abstract

Hexagonal boron nitride (h-BN) has recently gained much attention due to its high thermal conductivity and low electrical conductivity. In this study, we proposed to evaluate the impact of the modification of h-BN for use in a polymethylmethacrylate/polyamide 6 (PMMA/PA6) polymer blend. Different methods to modify h-BN particles and improve their affinity with polymers were proposed. The modification was performed in two steps: (1) a hydroxylation step for which three different routes were used: calcination, acidic treatment, and ball milling using gallic acid; (2) a grafting step for which four different silane agents were used, carrying different molecular or macromolecular groups: the octadecyl group (Si-C18), propyl amine group (Si-NH2), polystyrene chain (Si-PS), and PMMA chain (Si-PMMA). The modified h-BN samples after hydroxylation and functionalization were characterized by FTIR and TGA. Py-GC/MS was also used to prove the successful graft with Si-C18 groups. Sedimentation tests and multiple light scattering were performed to assess the surface modification of h-BN. Granulometry and SEM observations were performed to evaluate the particle size distribution after hydroxylation. After the addition of Si-PMMA modified h-BN into a PMMA/PA6 co-continuous blend, the morphology of the polymer blend nanocomposites was characterized using SEM. The calculation of the wetting parameter based on the surface tension measurement using the liquid drop model showed that h-BN dispersed in the PA6 phase. Grafting PMMA chains onto hydroxylated h-BN particles combined with an adequate sequence mixing led to a successful localization of the grafted h-BN particles at the interface of the PMMA/PA6 blend.

## 1. Introduction

Polymeric materials make the perfect candidate for application in electronics due to their properties such as their electrical insulation, being lightweight and easy to process, and their low cost [1]. As one of the most quickly advancing fields, electronic devices are continually being improved to enable better performances all while maintaining or miniaturizing their size and weight. A key factor for enabling this is the ability to control heat management, as thermal accumulation can lead to the deterioration of the operating speed, efficiency, and reliability performances of the devices [1,2,3]. While polymers are known for their low thermal conductivity (0.1–0.5 W·m^−1^·K^−1^) due to their organic nature [4,5,6], thermally conductive fillers provide a way to transfer heat by improving the thermal conductivity of the polymers [7,8,9,10,11,12]. One of the most promising fillers for modulating the thermal conductivity of polymers is hexagonal boron nitride (h-BN). This material is known for its unique properties, combining electrical insulation and thermal conduction. h-BN is also called white graphite due to its color and its atomic-scale hexagonal structure, which is close to that of graphite. B–N bonds are partially ionic due to the higher electronegativity of the N atom, which differs from the purely covalent C–C bonds of the graphitic structure. Van der Waals-type interlayer interactions make h-BN more difficult to exfoliate and functionalize than graphite [13]. h-BN is widely used for cosmetics, electrical insulation, lubrication, and for its microwave-transparent properties. Due to its layered structure, h-BN exhibits an anisotropic thermal conductivity with values in-plane of 300 to 600 W·m^−1^·K^−1^ and through-plane of 2 to 30 W·m^−1^·K^−1^ [4,13,14]. In attempts to improve thermal conductivity, various experimental strategies have been developed and implemented for the incorporation of h-BN platelets or its exfoliated nano sheets (BNNS) into polymer matrices. It is reported that many factors, including the filler structure, orientation, content, shape, and aspect ratio, affect the thermal conductivity of such composites [12,15,16]. In addition, the processing parameters play a role in the final properties of the polymer composites. A very high filler content generally leads to an increase in melt viscosity, which can be detrimental to thermal conductivity because the higher the viscosity, the greater the porosity of the final material [17,18]. Moreover, the thermal conductivity of composites is expected to be higher when the concentration of conductive fillers reaches the percolation threshold. Thus, controlling the particle organization within the polymer matrix is crucial. Some strategies rely on simply embedding h-BN particles in thermoplastics by melt compounding [19]. Other approaches consist of controlling the material morphology by localizing h-BN sheets in a continuous network and controlling their orientation [20]. Only a few of these studies achieved the targeted morphology, demonstrating that substantial efforts are still required to improve the thermal properties of h-BN composites.

The main issue is to reach a suitable dispersion state of fillers within the polymer matrix. Another important factor is to maintain a strong matrix–filler interfacial adhesion. Indeed, a strong adhesion ensures a continuous heat transfer and therefore impedes the formation of an interfacial thermal resistance [4].

Both dispersion state and particle/matrix interfacial adhesion are driven by the filler–matrix chemical bonds. While pristine h-BN exhibits a limited dispersion in polymers due to poor affinity, surface modification is considered to be an effective strategy to reduce particle aggregation and to favor homogenous dispersion in composites matrices. Furthermore, the percolation threshold is reached at a lower concentration when the particle size is smaller, explaining why many studies aim to exfoliate boron nitride aggregates.

Chemical functionalization is one attractive method for improving interfacial interactions. It requires the presence of chemically active functional groups on the h-BN sheet edges such as hydroxyl -OH, alkoxy -OR, amine -NH_2_, and alkyl -R groups [14,21]. The latter can further react with a foreign molecule bearing the adequate chemical function. Among the different functionalization routes, hydroxylation is the most important method prior to the modification of h-BN materials [22,23]. Hydroxylation is carried out to introduce -OH groups on bore atoms at the edges of h-BN sheets; this could be easily carried out by methods such as direct oxidization in the air [24,25] and sonication in various solvents [26,27]. The introduction of these -OH groups at the edges of h-BN promotes its homogeneous dispersion in solutions and also improves its affinity with polymer matrices. Cui et al. [25] reported the efficient exfoliation and functionalization of boron nitride via a thermal oxidization process. His work was an attempt to realize a similar protocol of hydroxylated graphite oxide (GO) sheets that enables the aqueous suspension of single sheets in high yield [28]. However, h-BN particles are more difficult to oxidize due to the lack of active sites onto the sheet’s surface, but only on the edges of the sheets. In addition, when comparing with graphite, it is more difficult to exfoliate due to strong ‘lip–lip’ interactions between neighboring nanosheet layers. Therefore, the chemistry routes used to modify graphite are not always efficient for h-BN. However, after thermal treatment, Cui et al. demonstrated a large amount of hydroxyl groups on the edges of h-BN.

Treatments using strong acids and bases were also reported in numerous works [29,30,31]. Wu et al. [32] demonstrated the hydroxylation of h-BN by sonicating h-BN particles in a nitric acid solution. Korycki et al. [33] compared four treatments to select the one that increased the hydrophilicity of h-BN platelets most significantly and therefore resulted in the highest concentration of the -OH groups grafted on sheet edges. As the results were proximate, the authors selected a thermal treatment that was more environmentally friendly than the use of strong acids or bases. Ball milling is another promising option for the modification of h-BN particles [29,34,35,36,37]. Ding et al. [38] demonstrated the synthesis of BNNS with an average thickness of about 2.0 nm and an extremely high production yield (~98%), as well as highly dispersed particles in water at high concentration (~35 mg·mL^−1^). However, it is worth mentioning that most studies showing the reduction and enlargement of the (002) peak in XRD have interpreted it as an exfoliation phenomenon of h-BN (and also graphite) platelets [10,39,40,41], which is not exactly the case. Indeed, this change in the (002) peak is instead attributed to a breakage of the primary particles leading to the same interlayer spacing, resulting in a lower number of platelets per particle. Exfoliation should be defined as the spacing of the particles with a distancing between the platelets, such as in clay nanocomposites [42]. Ding et al. [38] mixed pristine h-BN with furic acid and grinded the particles via ceramic balls of different sizes. This option led h-BN edges to functionalize with -OH groups and simultaneously to separate nanosheets via interaction with aromatic groups of furic acid.

For further covalent functionalization, silane treatments have been employed on h-BN particles bearing hydroxyl groups on their edges [32,43,44,45,46,47,48,49,50]. After hydrolysis of the alkoxysilane groups, the obtained silanols reacted with the hydroxyl groups on the h-BN edges. The chemical structure of the silane agent was chosen to present a chemical affinity with either the polymer matrix or the solvent, allowing for a better dispersion either in the organic or aqueous phase. Yu et al. [45] prepared h-BN/epoxy resin nanocomposites via the sol–gel method, starting with the hydroxylation of h-BN via a thermal treatment and using (3-isocyanatopropyl)triethoxysilane as a coupling agent, which enhanced the thermal oxidative stability. As a result, the functionalization of h-BN with this organosilane allowed the uniform dispersion of the particles into epoxy resin, as proven by TEM.

In most cases, conventional thermally conductive h-BN/polymer composites need the incorporation of high quantities of boron nitride particles, up to 60 wt%, to obtain high thermal conductivity (TC) (>5 W·m^−1^·K^−1^) [51]. However, such high filler loading leads to laborious processing due to the high viscosity and prohibitive cost for the obtained polymer composites. A few studies proposed an ingenious morphology by aligning boron nitride sheets within a polymer matrix to obtain a high TC with lower amounts of h-BN [40,41,52]. For increasingly significant TC of polymer composites, an effective strategy consists of forming percolated filler networks within the polymer. When reaching the percolation threshold, the heat transfer will be more efficient while the interfacial thermal resistance will be minimized [20,24,41,53,54,55,56,57,58,59,60]. This morphology can be obtained by dispersing the fillers in one of the two co-continuous phases of an immiscible polymer blend. This was clearly demonstrated for graphene [61], carbon black [62], and carbon nanotubes [63]. A few recent studies demonstrated the localization of boron nitride particles in co-continuous polymer blends. Jian et al. [59] reported the construction of a double percolation structure by controlling the distribution of modified h-BN in the polystyrene phase in a PS/PP blend. The polymer composite achieved a high thermal conductivity of 0.45 W·m^−1^·K^−1^ at 14.5 wt% loading of modified h-BN. Cao et al. [20] investigated the selective localization of h-BN in PP/EPDM blends from thermodynamics and the kinetic aspects of different blending processes. The thermal conductivity of (h-BN/PP)/EPDM composites (BN being pre-dispersed in PP and EPDM added afterwards) was higher than that of single matrix h-BN/PP composites. With h-BN particles dispersed in PP phase at 40 wt%, the thermal conductivity of (h-BN/PP)/EPDM composites reached a high value of 1.37 W·m^−1^·K^−1^. The majority of works based on boron nitride selective localization reached only a relatively weak thermal conductivity, and only a few studies report high thermal conductivity, such as the work done by Liu et al. [54] where they prepared a ternary polystyrene/polypropylene/boron nitride (PS/PP/h-BN) composite with 3D-segregated filler networks by the solution-mixing and hot-pressing methods. The authors reported that the ternary composite containing 50 wt% h-BN achieved a thermal conductivity of 5.57 W·m^−1^·K^−1^. Even though such morphology enables the composite to reach a high TC, it still requires a large amount of h-BN since the particles are localized inside one phase of the ternary polymer blend.

In contrast, in our study, we selectively localized boron nitride at the interface of two phases into a polymer blend to reach the percolation threshold at low h-BN content. We selected PMMA/PA6 as the immiscible thermoplastic blend, giving rise to co-continuous morphology. To date, this strategy has not been explored to obtain thermally conductive composites. To achieve such controlled morphology, boron nitride particles were modified via a two-step procedure. In the first step, they were hydroxylated via three different methods. In the second step, four well-chosen different silane agents were used for the grafting of the hydroxylated particles. These modifications were assessed using FTIR, TGA, Py-GC/MS, SEM, sedimentation, and granulometry analyses. Finally, grafted h-BN was introduced into a PMMA/PA6 polymer blend to investigate the interfacial localization of the modified particles. The combination of the grafting and a sequence mixing led to the successful localization of grafted h-BN particles at the interface of the PMMA/PA6 blend.

## 2. Materials and Methods

### 2.1. Materials

Hexagonal boron nitride (h-BN) particles (3M Cooling Filler Platelets CFP003F, St Paul, MN, USA) with an average size of 2–6 µm (d_0.5_) and a surface area of less than 20 m^2^.g ^−1^ were provided by 3M Technical Ceramics (Kempten, Germany). Nitric acid solution (HNO_3_, 68% *w*/*w*) was supplied by Prolabo (Paris, France), and gallic acid powder with purity of 98% was purchased from Acros Organics, Geel, Belgium. Silane grafting agents (Figure 1) trimethoxyoctadecylsilane (named Si-C18) and 3-aminopropyletriethoxysilane (named Si-NH_2_) were purchased from ABCR (Lyon, France), while poly(methylmethacrylate-co-3-(triethoxysilyl)propyl methacrylate) 95/5% mol noted Si-PMMA and poly(styrene-co-3-(triethoxysilyl)propyl methacrylate) 95/5% mol noted Si-PS were synthesized in the laboratory using a protocol detailed in a previous study [64]. Polymethylmethacrylate (PMMA) Altuglas V825T was purchased from Arkema (Milan, Italy), and polyamide (PA6) Ultramid B33 was purchased from BASF (Ludwigshafen, Germany). All chemicals were used as received without any purification.

### 2.2. Methods

#### 2.2.1. Thermal Treatment

4 g of boron nitride particles h-BN were calcinated in a furnace at 1000 °C for two hours under air. The furnace was heated to 1000 °C at a heating rate of 10 °C/min and maintained at that temperature for 1 h 30; the cooling step was done by switching off the furnace. Hydroxylated boron nitride BNO was then collected. The resulting BNO^Cal^ was washed three times with deionized water and centrifuged to remove any boric acid formed during calcination treatment. The washed sample was named BNO^Cal/W^.

#### 2.2.2. Acidic Treatment with Nitric Acid

600 mg of h-BN powder was dispersed in 350 mL of HNO_3_ (68 wt%) and sonicated for 7 h using an Elmasonic sonication bath S100H (Singen, Germany) with an output power of 550 W. The hydroxylated boron nitride was repeatedly washed with water and centrifuged until obtaining a neutral pH = 7. The sample was noted as BNO^HNO3^.

#### 2.2.3. Ball Milling Treatment Using Gallic Acid

4 g of h-BN was mixed with 20 g of gallic acid and ceramic balls (sizes of 9 mm, 2–2.5 mm, and 0.6–0.8 mm). The mixture was milled for 2 h 30 min at 80 rounds per minute using a Retsch S1000 ball mill apparatus (Haan, Germany). The obtained boron nitride was then washed with deionized water and filtered. The obtained sample was dried using an Alpha 1–2 LDplus freeze dryer (Osterode am Harz, Germany) in order to eliminate water and to limit the agglomeration of the h-BN. The sample was noted as BNO^BM^.

#### 2.2.4. General Procedure for the Grafting of BNO

2 g of modified h-BN (either BNO^Cal^ or BNO^BM^) was mixed with 7 wt% of silane agents (either Si-C18, Si-NH_2_, Si-PMMA, or Si-PS) in 100 mL of an ethanol/water (90/10 wt%) solution (for Si-C18 and Si-NH_2_) or toluene (for Si-PMMA or Si-PS). The solution was then heated at solvent reflux for 15 h under stirring. After reaction, the product was centrifuged to eliminate the solvent and then washed two times with ethanol (or toluene) and two times with acetone. The obtained product was dried under a vacuum before characterization. The different samples obtained are listed in Table 1.

#### 2.2.5. Melt Extrusion of PMMA/PA6/BN Nanocomposites

The dispersion of grafted h-BN in the PMMA/PA6 polymer blend was realized using a melt blending process via a microcompounder DSM at 250 °C, 80 rpm and for 4 min. The polymers and the h-BN were vacuum dried at 80 °C overnight. A filler of 8 wt% was used with the 50/50 PMMA/PA6 blend. The different materials were introduced into the microcompounder simultaneously. To orientate the h-BN platelets that were hydroxylated by calcination and then grafted with Si-PMMA, the sequence of mixing was changed: the modified h-BN was first dispersed in toluene that contained the dissolved PMMA granules. The mixture was then cast, and the film obtained after the solvent’s evaporation was introduced in the microcompounder before the PA6 granules were added. The extrusion parameters were the same as those mentioned above.

#### 2.2.6. Characterization of the Modified h-BN Samples

A Fourier transform infrared (FTIR) analysis was performed using a Vertex 70 spectrophotometer (Bruker, Paris, France) in attenuated total reflectance (ATR) mode in the 4000–400 cm^−1^. The resolution was 4 cm^−1^, and 32 scans were accumulated for an improved signal-to-noise ratio. A thermogravimetric analysis (TGA) was performed on a PerkinElmer TGA8000 (Groningen, The Netherlands) at a heating rate of 10 °C/min under nitrogen atmosphere, and the temperature range was from 110 to 950 °C, with a previous isothermal step at 110 °C for 20 min. The TGA curves shown started after an isothermal step at 110 °C that allowed for the elimination of physisorbed water.

Raw, hydroxylated, and grafted h-BN were dispersed in water to assess the sedimentation phenomenon across time and to characterize the surface modification of h-BN. We dispersed 0.15 g of raw or modified h-BN into 25 mL of water (concentration of 0.6 wt%). The suspensions were sonicated with an Elmasonic bath for 1 min to help the dispersion of particles and were kept for 14 days at room temperature. The kinetics of settling at room temperature was also studied using the multiple light scattering analyzer TurbiscanLab from Formulaction (Toulouse, France). Each sample was introduced in a 25 mL glass (0.075 g of particles in 15 mL of water) cylindrical cell and scanned using a light beam. The concentration of h-BN in water was 0.5 wt%, and the suspensions were sonicated for 1 min for each sample. The analysis duration was divided into 3 steps to thoroughly follow the settling phenomena while avoiding useless data files. In the first step, when the settling was fast, a scan was recorded every 25 s for 1 h and 10 min. In the second step, a scan was recorded every 10 min for 2 h and 30 min, and in the last step, a scan was recorded every 30 min for 20 h. The total scanning duration for each sample was 23 h and 40 min. The backscattered light was normalized by the initial intensity such that ΔR (%) = R(t)/Ri, with R(t) representing the intensity registered at each time lap and Ri representing the intensity at t =0. The particle sizes were analyzed by a laser diffraction granulometer LSI3320 (Beckman Coulter, Brea, CA, USA)). The conditions were the same as they were for the sedimentation tests in water: 0.5 wt% of particles in water.

#### 2.2.7. Py-GC/MS

A Pyroprobe 5000 pyrolyzer (CDS analytical, Oxford, PA, USA) was used to pyrolyze the different h-BN samples under helium. This pyrolyzer used an electrically heated platinum filament. The samples (<1 mg) were introduced in a quartz tube between two pieces of rockwool and were pyrolyzed using a coil probe. The samples were heated at 600 °C for 15 s, and the gases were then drawn by the GC for 5 min. The pyrolyzer was interfaced to a 450-GC chromatograph (Varian) by means of a transfer line heated at 270 °C. In the GC apparatus, the initial temperature of 70 °C was raised to 310 °C at 10 °C/min. The column used was a Varian VF-5ms capillary column (Paris, France) (30 m × 0.25 mm; thickness of ¼ 0.25 μm). Helium was used as the carrier gas (1 l/min), and a split ratio of 1:50 was chosen for the analyses. The gases were introduced from the GC to the 240-MS mass spectrometer (Varian, Paris, France) through the direct-coupled capillary column.

#### 2.2.8. SEM

A scanning electron microscopy observation was performed using Quanta FEI 200 (Czech Republic). Cross-sections of the composite samples were prepared using cryo-fracture or polishing. For some micrographs, the PMMA phase was solubilized using toluene.

#### 2.2.9. Contact Angle Measurement

Contact angle measurements were carried out by depositing a liquid drop with controlled volume on the sample surface. The contact angle θ between the liquid and the substrate was measured using a drop shape analyzer DSA30 Kruss goniometer apparatus (Kehl, Germany) equipped with a CCD camera. The contact angles were measured on thin, flat, round disks of 1.5 mm thickness and of 25 mm diameter. Pure polymer disks were prepared using injection molding with a Zamak Mecrator Mini-press (Skawina, Poland) at 250 °C for 3 min with a constant pressure of 5 bars. h-BN and BNO^Cal^/Si-PMMA disks were made by pressure molding at room temperature using Struers Prontopress (Copenhagen, Denmark) at 30 bars for 4 min. Contact angle measurements between the sample flat surface (polymers or compacted h-BN) and the two solvents (water and diiodomethane) with different dispersive $\gamma_{L}^{d}$ and polar $\gamma_{L}^{p}$ contributions were then performed three times for each sample.

#### 2.2.10. Solvent Etching

Solvent extraction was used to further evaluate the localization of h-BN particles in the polymer blend. We immersed 150 mg of cylindrical shape samples in 20 mL of toluene and heated them at 65 °C for 48 h to selectively dissolve the PMMA phase. The etched solid phase was then washed with fresh toluene and acetone and dried for 24 h.

#### 2.2.11. Thermal Diffusivity Measurements

Thermal diffusivity experiments were conducted on disks of 25 mm diameter and 1.5 mm thickness using a Xenon flash analyzer (Linseis XFA 600, Germany). The thermal diffusivity was calculated through the thickness of the sample by measuring the time required for the temperature rise. Each sample was measured, and the average was taken, along with the standard deviation.

## 3. Results and Discussion

### 3.1. Hydroxylation of h-BN (TGA and FTIR)

Three hydroxylation treatment methods were performed on h-BN particles in order to introduce -OH groups on the edges of the platelets (Figure 2). The first method was a heat treatment at 1000 °C. The second treatment used a ball milling step in the presence of gallic acid. The third procedure combined nitric acid and sonication. Figure 3a shows the FTIR spectra of BNO^Cal^ and BNO^Cal/W^ compared with the raw h-BN. Prior to heating, the spectrum of pristine h-BN exhibited only the characteristic peaks of B-N-B out of the plane bending observed at 775 cm^−1^ and B-N in-plane stretching at 1313 cm^−1^. A huge band appeared at 3202 cm^−1^ for BNO^Cal^ that was attributed to the presence of boric acid formed during the calcination. The formation of boric acid during the thermal treatment of boron nitride was already described [25]. This was confirmed by a TGA (Figure 3b) showing the thermal degradation of boric acid for BNO^Cal^ between 150 °C and 300 °C, which is in accordance with the literature [25,45].

After washing, the obtained BNO^Cal/W^ sample showed disappearance of the boric acid bands, even at 3200 cm^−1^ (Figure 3a). However, due to the very low amount of hydroxyl groups compared with the size of the h-BN sheets, the presence of those hydroxyl groups were not visible with FTIR. This is in accordance with the literature [33]. The TGA (Figure 3b and Figure 4b) of BNO^Cal/W^ also showed a weak weight loss in comparison to h-BN that confirmed the difficulty of assessing a high hydroxylation rate.

It is well-known that characteristic bands of hydroxyl groups formed on h-BN platelets’ edges can appear in the FTIR spectrum between 2900 and 3500 cm^−1^ [25,45]. After hydroxylation by ball milling (BNO^BM^), a weak band around 3455 cm^−1^ was seen, which was attributed to the -OH groups of the h-BN sheet edges, as shown in Figure 4a. Additionally, in Figure 4b, a steady weight loss was also visible between 200 °C and 900 °C, which corresponded to the removal of the -OH group. These results confirm that hydroxylation using the ball milling method is more efficient than that of calcination. The last hydroxylation method with the sonication of h-BN in nitric acid (BNO^HNO^_3_) also demonstrated a successful hydroxylation of h-BN. Indeed, the FTIR showed the clear appearance of bands around 2900 cm^−1^ for the ball milling treatment characteristics of the -OH groups. These signals were more intense, which agreed with the TGA results. Indeed, the weight loss was significantly higher for BNO^HNO3^, with a value of about 1% loss up to 500 °C. Moreover, there were two stages of thermal degradation in the TGA for BNO^HNO3^: the first one started around 200 °C and was probably due to edge-hydroxylation, whereas the second one (400 °C) was due to in-plane hydroxylation [65].

To conclude, the TGA and FTIR measurements confirmed the efficiency of the hydroxylation of h-BN for the ball milling and nitric acid methods. Even if the quantification was difficult based only on FTIR and TGA, the best results were obtained using the nitric acid treatment coupled with sonication. The calcination method showed the lowest weight loss, but it held the simplest procedure to be performed. Moreover, this method treated a larger quantity of material without using solvent when compared with other methods.

#### 3.1.1. Stability of the Suspensions of Hydroxylated h-BN in Water

The stability of the suspensions of solid particles in a liquid media is mainly governed by these parameters: the density of each phase, viscosity of the liquid, temperature, particle size and shape factors, and particle concentration, as well as the particle–liquid interaction and particle–particle interaction. Suspending solid particles in a liquid is a relevant method for testing surface treatment modification. Indeed, as the surface treatment affects the surface chemistry, the particle–liquid and interparticle interactions are modified, resulting in the slowing down or speeding up of the settling kinetics. Water was chosen by many authors as the dispersion medium because of sustainability reasons. Most importantly, depending on the grafted molecules, the high polarity of water makes it possible to interact with polar groups of the grafted part. In the case of boron nitride, several studies demonstrate the settling variations in water according to the surface modifications [33].

In our study, we aimed to prove the modification of h-BN sheets by the introduction of hydroxyl groups. The stability of the particles was expected to be longer when the hydroxyl groups density was higher, stemming from more -OH–water interactions. For this purpose, sedimentation tests were carried out by taking pictures of low concentrated suspensions each day for 15 days. As seen in Figure 5a, the raw h-BN started to settle down within the first day. After 8 days, almost the whole sediment was formed at the bottom of the vial, and by 15 days, the middle part of the vial became even clearer. Some raw h-BN particles stayed in suspension even after 15 days in water, which we assumed could be attributed to the smallest particles. Indeed, the size distribution presented in granulometry results below of hydroxylated BNO indicates a part of particles whose size was below 1 micron, for which the gravity force was too weak to fight the thermal agitation kT.

For all the treated h-BN, the sedimentation kinetics was slower, but the difference between the three treatments was not obvious. Nevertheless, the following trends were noticed: The sediment height was much thinner than that of h-BN, and the middle part of the vials was more opaque than that of the h-BN one, even after 15 days. The multiple light scattering experiments (next section) were performed to go further into the characterization of the stability of suspension.

Boron nitride hydroxylated by sonication in nitric acidic (BNO^HNO3^) showed the thinnest sediment at the bottom of the vial: the particles were more stacked, which could indicate that the interparticle interactions were weak or at short range. In the case of BNO^Cal/W^, a supernatant phase was present, stemming from the capillary bridges formed between the hydrophobic particles entrapped in air bubbles. This phenomenon is generally related to hydrophobic particles, which have more affinity with air than water.

The better stability of the treated particles compared with that of raw h-BN was explained by the hydroxyl groups formed at the edge of the h-BN sheets. The second hypothesis was a possible exfoliation or otherwise breakage of the primary particles of h-BN sheets during the treatments, specifically during the ball milling process [38]. These particles could be more stable in water suspension.

The Turbiscan Stability Index (TSI) is a parameter used to characterize the global stability of suspensions. Its calculation is based on an integrated algorithm that sums up the evolution of transmitted or backscattered light at every position measured (h), based on a scan-to-scan difference, over total sample height (H). It investigates the stability of suspensions containing the h-BN particles in water by the evaluation of their sedimentation kinetics. Figure 6 shows the evolution of the global TSI for the different h-BN samples in water. During the first minutes, the BNO^HNO3^ particles destabilized very rapidly compared with the other samples. Over time, all the hydroxylated samples exhibited a TSI curve below the one of pure h-BN, which corresponded to a better stability in water than that of raw h-BN. This proved an evolution of the affinity of the h-BN particles with water after the use of the three hydroxylation methods. After more time, the BNO^HNO3^ particles seemed to be slightly more stable than the calcinated and ball-milled particles. More insight on the stability of water suspensions by Turbiscan is shown in Appendix A. This result agrees with the TGA and FTIR results. Indeed, BNO^HNO3^ showed the highest weight loss in the TGA and the highest intensity for the OH band in FTIR.

#### 3.1.2. Hydroxylated h-BN Particle Size Measurements

The microstructure of h-BN and hydroxylated BNO particles was observed using SEM analysis (Figure 7), and their particle diameters were measured to study the effect of hydroxylation using the three methods on the size distribution and agglomeration. Upon the initial observation of the SEM images, we noticed a relative agglomeration for calcinated particles, as well as some type of wear in the edges of some sheets (Figure 7b). Ball-milled particles demonstrated the most changes in particle structure, where we noticed different sizes with especially smaller particles (Figure 7c); this was the result of the ball milling cleavage of h-BN platelets. One other thing we noticed was the wear of the platelets’ edges, where it seemed that ball milling also resulted in some damage on the platelets’ structure, especially on the edges. Sonicated h-BN demonstrated a more similar structure or distribution than the original h-BN particles.

To evaluate the size of the h-BN particles, we measured from these SEM images the diameters of 100 platelets for each sample (Figure 8). Most particles of h-BN were in the range of 0.1 to 0.5 µm, and a small number of platelets had a diameter greater than 1 µm. As expected, there was a high difference between the d_0.5_ given by the supplier, obtained by laser light scattering in ethanol (2–6 µm), and the sizes obtained here by SEM. The calcinated samples showed fewer particles of 0.1 µm and more particles between 0.2 and 0.6 µm. Ball milling demonstrated the biggest reduction in size, where more than half of the particles measured showed a diameter less than 0.2 µm; the second most noticeable diameter was 0.4 µm. The HNO_3_ treatment oddly displayed smaller particles at 0.3–0.5 µm than did the h-BN sample.

To further study the overall h-BN particle size distribution and to explain the sedimentation observations, we performed particle size measurements using laser granulometry. The results in Figure 9 display the particle distribution associated with the volume occupied by the particles. Hence, the biggest particles tended to increase the final mean volume diameter, explaining the difference between the diameter reported in Figure 9 and the diameter reported in Figure 8. The h-BN particles demonstrated a single peak with a general volume diameter of 15 µm, while the calcinated BNO^Cal/W^ particles displayed one peak with a bigger average size of 17.83 µm, which could be the result of the moderate agglomeration of particles after thermal treatment and washing. This agglomeration was also noticed in some SEM images. The h-BN sonicated in HNO_3_ showed an average volume size of 10.68 µm. The BNO^BM^ showed a different behavior, with three peaks centered on the diameter values at 0.25 µm, 1.41 µm, and 4.65 µm. This confirmed the formation of many small fractions upon ball milling. Further analyses on size distribution are provided in the Appendix A.

### 3.2. Grafting of Calcinated BNO^Cal/W^ with Four Different Silane-Grafting Agents

As a reminder, calcination, ball milling, and nitric acid treatments are efficient methods of introducing -OH groups onto h-BN sheets. We attempted to modify the surface chemistry of h-BN sheets by grafting silane agents from these -OH groups. Next, we present our study of the grafting of the particles obtained by calcination. Indeed, the calcination procedure combined the simplicity of the hydroxylation method and the satisfaction of the hydroxylation rate. Grafting ball milled h-BN particles was also realized and is demonstrated in the Appendix A. The silane grafting was successful for both hydroxylation methods: the ball milling and calcination.

Four different silane agents were grafted on hydroxylated BNO^Cal/W^ particles using the same protocol detailed in the experimental section. The silane-grafting agents (Si-C18, Si-NH_2_, Si-PMMA, and Si-PS) were chosen to have different affinities with the polymer matrices. Si-C18 and Si-NH_2_ are classical silane agents used to modify particles in order to improve their affinity with polyolefins and polar polymers, respectively. Si-PMMA and Si-PS are copolymers previously synthesized in our lab. When grafted on particles such as h-BN, they can improve their affinity with PMMA and PS polymers, respectively. This type of graft can target the localization of Si-PMMA-grafted and Si-PS-grafted BNO^Cal/W^ in PMMA and PS polymers when they are blended with other matrices such as polyamide. The four grafting procedures proved to be successful on BNO^Cal/W^. The presence of the four grafting agents on BNO^Cal/W^ particles was confirmed by TGA, FTIR, and py-GC/MS, as well as by sedimentation tests. The FTIR analyses are presented in Figure 10. Bands corresponding to the CH_2_ stretching groups were present in the zone between 2800 and 3000 cm^−1^ for all grafted samples. BNO^Cal/W^/Si-PS also showed a band at 699 cm^−1^ that corresponded to the CH-bending groups of monosubstituted aromatic groups of styrene units, while the BNO^Cal/W^/Si-PMMA spectrum showed a peak band at 1734 cm^−1^ that corresponded to the C=O groups of the methacrylate units.

In the TGA thermograms of the modified samples (Figure 11a), a significant weight loss was noted for the BNO^Cal/W^ grafted with the four grafting agents in comparison to the BNO^Cal/W^ itself. The weight loss was mostly centered around 350–450 °C, which was the expected degradation temperature for the organic molecules and macromolecules grafted on the BNO^Cal/W^ particles. The grafting with the Si-NH_2_ groups exhibited the lowest weight loss with 1.8 wt%, which was in accordance with the FTIR analysis (Figure 10) that showed the weakest change in relation to the BNO^Cal/W^ spectrum. On the contrary, the grafting of Si-PS showed the highest weight loss of around 3 wt% for BNO^Cal/W^/Si-PS. It is noteworthy that the weight loss measured by the TGA was dependent on the molecular weight and the grafting rate of the silane.

The stability of the grafted h-BN particles was followed by sedimentation in water (Figure 5b). Whatever the nature of the grafted chains, a supernatant is visible in Figure 5b from the beginning of the observations. This upper layer looked thicker for BNO-C18 and BNO-NH_2_. After 8 days, the sedimentation looked almost stabilized for all the samples. The clearest suspension was obtained for BNO-NH_2_ but with large particles in suspension in a very clear liquid media. For the other samples, all the suspensions were opaque in the middle of the vial; the most opaque was the BNO-PMMA. As seen in Figure 5b, the sediment layer was thicker for raw h-BN, followed by BNO-NH_2_. Since the settling kinetics was a balance of several phenomena, a visual observation was not sufficient to confirm the particles–water affinity. Multiple light scattering was used to follow the evolution of the transmitted and backscattered light throughout the samples. The curves are shown in Appendix A. All the specimens showed similar trends, with a peak of transmitted light appearing at the top of the vial (h-BN, BNO-PMMA, BNO-PS) and at the center (BNO-C18). The backscattered light curve represented the opacity of the sample. The evolution of backscattered light was quite regular with time for h-BN, BNO-PMMA, and BNO-PS, indicating a continuous settling kinetics. On the contrary, the distribution of the particles was not homogeneous along the vial for BNO-C18; the particles were located in the lowest part of the vial. This was explained by the fast sedimentation of BNO-C18. In the lowest part, where the particle concentration was higher than that of the other specimens, the evolution was slow. Lastly, the evolution of backscattered light for BNO-NH_2_ was very fast in the first 10 min, with a large amplitude motion of particles. There was almost no change with time. It is worth noting that a supernatant was observed in the upper part for all these specimens, even if its thickness could not be measured when there was no change in that part of the vial. Due to the combination of several phenomena, the stability index (TSI) is preferred to compare the stability of the samples.

The stability index in Figure 11b indicated a very fast destabilization kinetics for BNO-NH_2_ during the first 400 min, and then its evolution was slower. BNO-PMMA and BNO-PS settled quite fast the first 100 min, and then the same slope was measured, indicating a similar trend for both. Only the BNO-C18 followed a regular increase from the beginning. According to TSI, the stability classification at 1400 min was as follows: BNO-NH_2_ < BNO < BNO-PMMA < BNO-PS < BNO-C18, with BNO-C18 being the most stable. Additional information can be seen in Appendix A.

Additional suspensions in toluene were prepared to assess the chemical change induced by the various grafting in a nonpolar solvent. The polarity of toluene was 0.1 relative to 1 for water. The data are presented in Appendix A. The TSI of suspensions in toluene is shown in Figure 11c. The destabilization was very fast for the three studied samples, and the evolution was weak after 400 min, contrary to the suspensions in water for which changes were visible after 1400 min. Since toluene is less polar than water, the particles with polar groups had less affinity with toluene, and they migrated faster to the bottom of the vial. Accordingly, the BNO-PMMA were the most stable particles in toluene. When comparing BNO-PS to h-BN particles, BNO-PS had less affinity with toluene than non-grafted h-BN, showing that BNO-PS particles are more hydrophilic than raw particles. Regardless of the treatment, the difference in settling kinetics proved that the boron nitride particles were successfully grafted. This grafting step was crucial to tune the surface tension of the particles in order to control their further localization in the polymer blends.

### 3.3. Dispersion of h-BN Particles in PMMA/PA6 Polymer Blends

Further applications of this study aimed to obtain highly thermally conductive polymeric materials with boron nitride particles. The idea was to use the co-continuous morphology of an immiscible polymer blend to control the localization of the boron nitride particles. The latter were expected to be situated into one phase or at the interface of the co-continuous morphology. This concept of double percolation was abundantly described in the literature with carbon-based materials (carbon black, graphene, and carbon nanotubes) [66]. To this day, only a few studies were published about boron nitride localization. Nonetheless, it has already shown promising and interesting results [20,54,59]. By achieving such particle organization, it is possible to dramatically decrease the percolation threshold, resulting in lightening materials for the same electrical or thermal conductivity. Both phenomena result in a percolation threshold created by the conductive particles. In our case, a PMMA/PA6 polymer blend was chosen because the conditions to obtain a co-continuous morphology are well-known [67,68].

Before introducing h-BN particles into PMMA/PA6 blends, contact angle measurement was carried out to determine the surface energy of the different components of the blends. The method is based on the measurement of the contact angle between a liquid drop and a solid surface by a goniometer. The interfacial tensions of the blend’s components are deduced from the contact angles through Young’s theory. Figure 12a,b show the drop shape pictures on h-BN and BNO^Cal/W^/Si-PMMA flat, thin disks: the two drops showed a fully different behavior. For the h-BN, an immediate absorption of the water drop was noticed, and a weak contact angle was registered at first contact between the water drop and h-BN surface. On the contrary, the boron nitride grafted with polymeric chains displayed a drastic change in its affinity with the water drop: it became significantly hydrophobic. In Figure 12b, the water drop made a very high contact angle after coming into contact with the BNO^Cal/W^/Si-PMMA surface and maintained the same shape after several minutes. This surface behavior shows the silane grafting from the localization of boron nitride particles can be controlled in PMMA/PA6 polymer blends.

Wetting parameter calculations: According to thermodynamics law, interfacial tensions between each component of a system containing PMMA, PA6, and h-BN govern the localization of h-BN fillers in the immiscible polymer blend. The determination of the wetting parameter (Ꞷ_AB_) is possible using the interfacial tensions between the components (Equation (1)) [69].
(1)$$\;{Ꞷ}_{\mathrm{AB}}=\frac{\gamma_{\mathrm{SB}}-\gamma_{\mathrm{SA}}}{\gamma_{\mathrm{AB}}}$$
(2)$$\gamma_{\mathrm{ij}}=\gamma_{i}+\gamma_{j}-\frac{4\gamma_{i}^{d}\gamma_{j}^{d}}{\gamma_{i}^{d}+\gamma_{j}^{d}}-\frac{4\gamma_{i}^{p}\gamma_{i}^{p}}{\gamma_{i}^{p}+\gamma_{i}^{p}}$$
(3)$$\gamma \mathrm{ij}\;=\;\gamma i\;+\;\gamma j-2\;\sqrt{\gamma_{i}^{d}\gamma_{j}^{d}}–\;2\;\sqrt{\gamma_{i}^{p}\gamma_{i}^{p}}$$

Interfacial tensions γ_ij_ between polymer (A), polymer (B), and solid filler particles (S) could be calculated using two different approaches depending on the type of component; the harmonic mean equation (Equation (2)) or geometric mean equation (Equation (3)) [70,71], where $\gamma_{i}^{d}$ and $\gamma_{i}^{p}$ are dispersive and polar contributions, respectively, to the total surface tension of the component γ_ij_, were calculated using Owens–Wendt’s and Wu’s methods. The wetting parameter of a solid particle (S) into a binary immiscible polymer blend (A and B) expresses the solid filler’s most favorable localization in order to minimize the blend’s free interfacial energy. As a result, the final localization of the fillers could be predicted based on the value of (Ꞷ_AB_), as demonstrated in Table 2 [72].

Table 3 summarizes the global surface tension values as well as the dispersive and polar contributions for each component, calculated from contact angles measured by the goniometer.

Table 4 shows the interfacial tensions as well as the wetting parameters calculated using the surface tensions reported in Table 3. The harmonic mean equation was used to measure the interfacial tension between polymers (PMMA/PA6), while the geometric mean equation was used for the polymer/filler interfacial tension. The results in Table 4 indicate that h-BN particles should be dispersed only in PA6 phase before grafting, while BNO^Cal/W^/Si-PMMA particles should be dispersed after grafting at the interface of PMMA/PA6 blend. However, we should note that these predictions remain theoretical and must be considered with caution due to the uncertainties concerning the surface energies of both polymers and boron nitride particles.

#### 3.3.1. Microstructure of PMMA/PA6/h-BN Blends

PMMA/PA6 blends were prepared by adding 8 wt% of boron nitride fillers to a 50/50 polymer blend. The various blends are presented in Table 5.

All the samples exhibited a co-continuous morphology in Figure 13a,b when we showed SEM images of a polished cross section of the samples. The blend containing non-treated boron nitride was formed by thin strongly elongated stripes characterized by a weak interfacial adhesion. As predicted by the wetting parameter, the h-BN particles were dispersed in PA6 phase. The blend with the PMMA chain grafted h-BN particles showed a different morphology with a coarser phase structure. Still, despite their functionalization, BNO^Cal/W^/Si-PMMA were located in the PA6 phase in Figure 13b. Even though the wetting parameter above 1 predicted their affinity with the PMMA phase, the modified h-BN particles were not hydrophobic enough to go into the PMMA phase. This could be explained by the value of the interfacial tension between PA6 and PMMA, which was rather low.

Selective solvent extraction was also used to examine the boron nitride particles in the polymer blend. The PMMA was dissolved, and during preparation, the solution containing the solubilized PMMA phase remained transparent. This observation confirmed that the boron nitride was fully located in the polyamide phase. The SEM image in Figure 13c clearly shows the h-BN agglomerates within the PA6 phase. When looking more closely at Figure 13e, one can see these 5 µm diameter agglomerates are composed of boron nitride layers. In contrast, in Figure 13d,f, such agglomerates are not visible, confirming that the BNO^Cal/W^/Si-PMMA-grafted particles were better dispersed in comparison to those of the non-grafted h-BN.

As a further attempt to localize the boron nitride particles at the interface of PMMA/PA6, the grafted BNO^cal/W^/Si-PMMA were mixed with PMMA by mixing the solvent in toluene. Next, the PMMA/BNO^cal/W^/Si-PMMA films were mixed with PA6 in a microcompounder. The SEM images in Figure 14a,c show that non-grafted h-BN particles migrated from PMMA phase to PA6 after melt mixing, while a great number (or the majority) of PMMA-grafted particles were localized at the interface of the two phases (Figure 14b,d).

#### 3.3.2. Thermal Diffusivity Measurements

Some thermal diffusivity experiments were conducted on disks of pure h-BN, PMMA/PA6, PMMA/PA6/BN, and PMMA/PA6/BNO^Cal/W^/Si-PMMA (Figure 15). It is noteworthy that this apparatus in most cases can only measure the through-plane thermal diffusivity for low thermal conductive materials such as polymers. Through-plane thermal diffusivity is expected to be lower than in-plane diffusivity because of the processing method. Indeed, the platelets are oriented in a planar direction due to shearing. The h-BN thermal diffusivity was found to be 0.512 mm^2^/s, and the PMMA/PA6 thermal diffusivity was 0.125 mm^2^/s. After introducing 8 wt% of h-BN, the thermal diffusivity reached a higher value of 0.141 mm^2^/s. However, there was no improvement when the h-BN particles were grafted with PMMA chains, which could be due to the extremely low percentage of fillers. Hence, the grafting could not show its effect on the thermal diffusivity of the ternary nanocomposite. An increasing of grafted h-BN fillers was made, amounting to 25 wt%, and the thermal diffusivity was improved to 0.211 mm^2^/s, almost double that of PMMA/PA6.

In spite of this improvement, further studies are in progress to ascertain a percolation of h-BN fillers localized at the interface by alternating multiple parameters, such as the proportion of the components in the composite, molecular weights of polymers, and sequence of mixing.

## 4. Conclusions

In conclusion, we compared different ways to hydroxylate h-BN platelets: ball milling led to particle breakage, whereas calcination promoted the formation of agglomerates in water. HNO_3_ leads to the highest hydroxylated rate. All hydroxylated h-BN were more stable in water than in h-BN, demonstrating a low but efficient hydroxylation. The grafting of the Si-C18 molecules onto hydroxylated h-BN was proven by TGA, FTIR, and Py-GC/MS. Various silane agents were successfully grafted onto the hydroxylated h-BN. The evolution of surface chemistry according to the nature of silane agents was confirmed by the surface tension measurement. We then tried to selectively disperse h-BN particles at the interface of a low interfacial tension blend (50/50 PMMA/PA6). The calculation of wetting parameter based on the surface tension measurement by liquid drop showed that the h-BN would disperse in the PA6 phase. The grafting of poly (methylemethacrylate-co-propyltriethoxysilane) 95/5% mol (named Si-PMMA) onto calcinated h-BN was successful, and the obtained PMMA-grafted h-BN was predicted to be localized at the interface of the PMMA/PA6 blend. This BNO^Cal/W^/Si-PMMA combined with an adequate sequence of mixing was proven to orientate the h-BN platelets at the interface of the co-continuous PMMA/PA6 blend. Despite the low thermal diffusivity of the blends with grafted boron nitride, these first results open the way to new, interesting studies investigating the influence of the interfacial localization of h-BN into a blend on the thermal conductivity.

## Figures and Tables

**Figure 1 nanomaterials-12-02735-f001:**
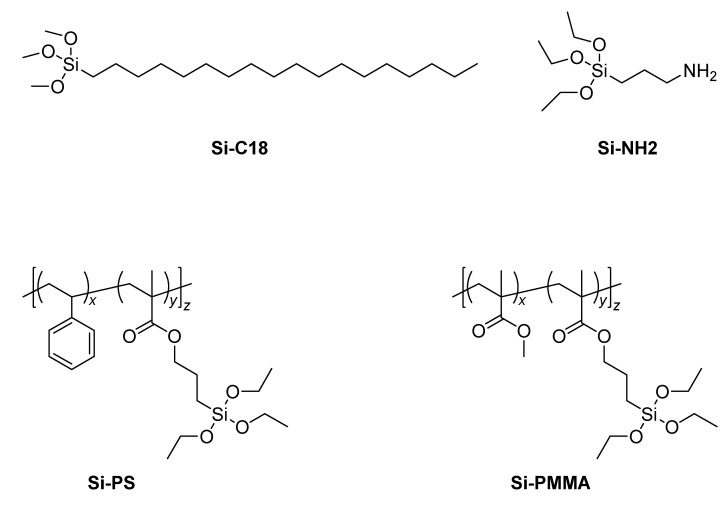
Schematic representation of the silane grafting agents used for the modification of hydroxylated h-BN.

**Figure 2 nanomaterials-12-02735-f002:**
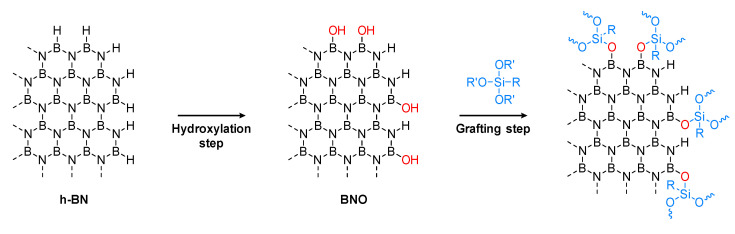
Schematic representation of the h-BN hydroxylation followed by a grafting step.

**Figure 3 nanomaterials-12-02735-f003:**
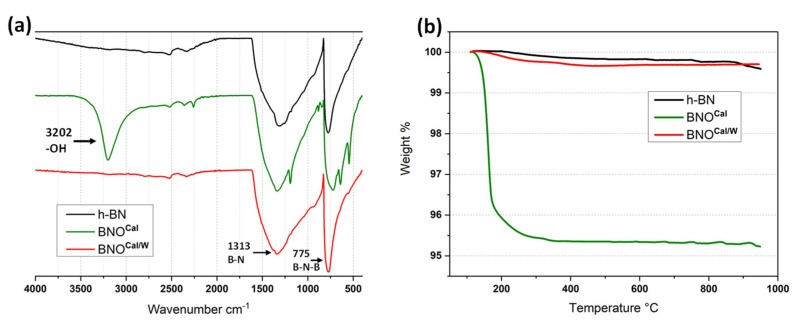
FTIR spectra (**a**) and TGA thermograms (**b**) of h-BN, BNO^Cal^, and BNO^Cal/W^ samples.

**Figure 4 nanomaterials-12-02735-f004:**
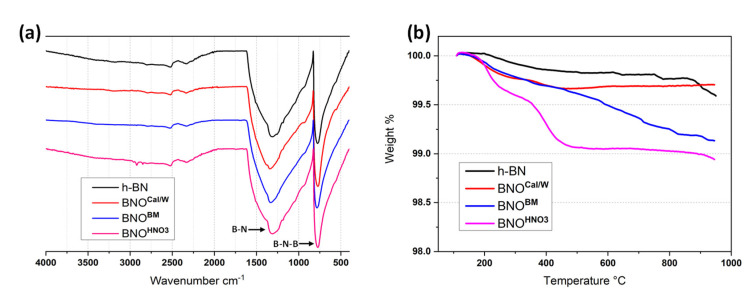
(**a**) FTIR spectra and (**b**) TGA thermograms of hydroxylated BNO with different methods.

**Figure 5 nanomaterials-12-02735-f005:**
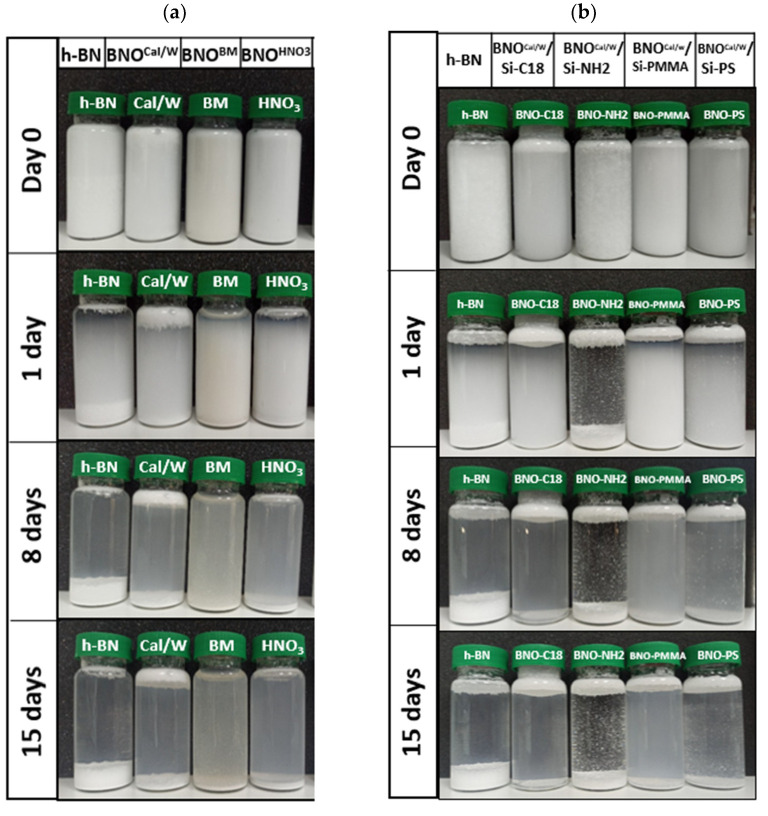
Photos of 0.6% wt suspensions in water. (**a**) Hydroxylated h-BN and (**b**) grafted BNO^Cal^.

**Figure 6 nanomaterials-12-02735-f006:**
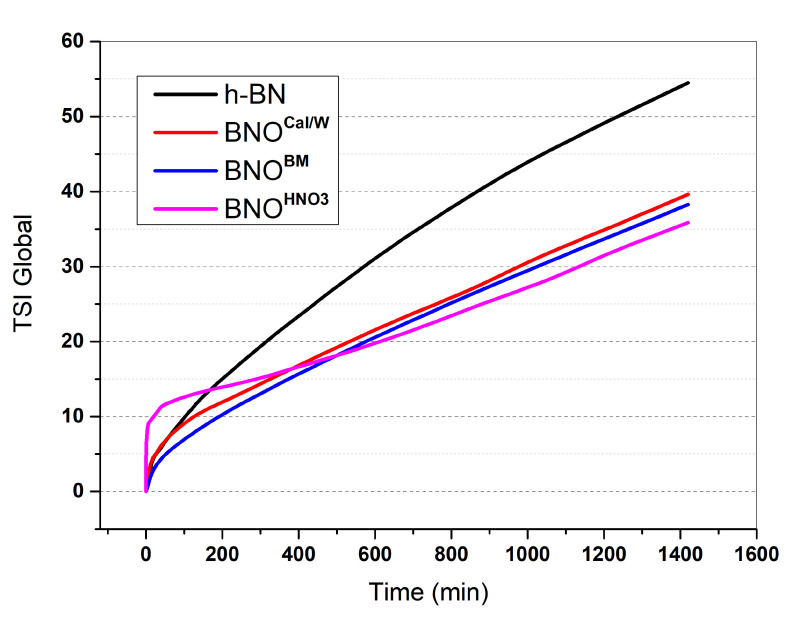
Turbiscan Stability Index (TSI) global stability of raw h-BN and hydroxylated boron nitride.

**Figure 7 nanomaterials-12-02735-f007:**
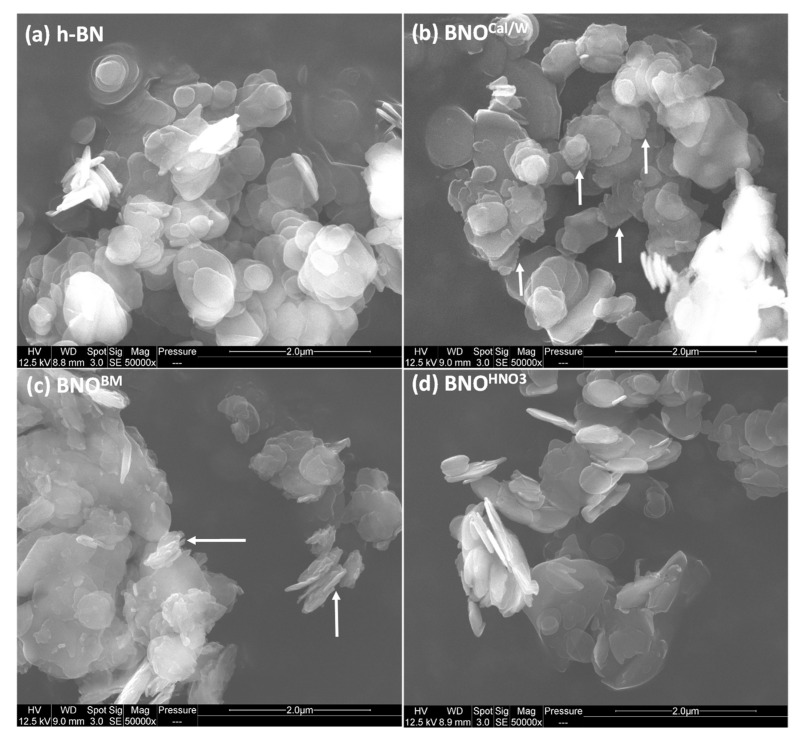
SEM images of (**a**) h-BN, (**b**) BNO^Cal/W^, (**c**) BNO^BM^, and (**d**) BNO^HNO3^ samples. Magnification: 50,000×, scale bar = 2 µm, and arrows indicate the evolution of the edges of the sheets.

**Figure 8 nanomaterials-12-02735-f008:**
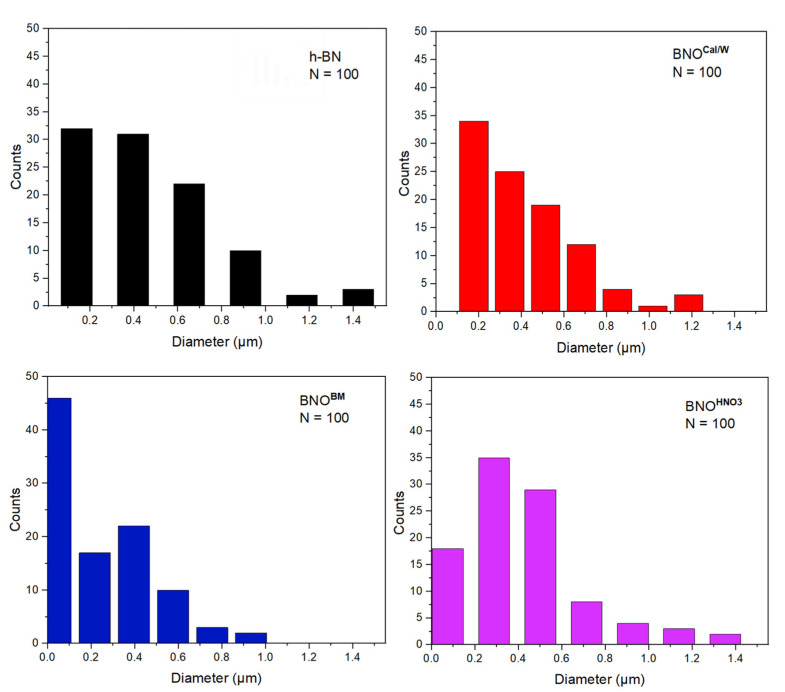
Particle size distribution of h-BN and hydroxylated BNO using ImageJ software (ImageJ; version 1.53s, Software for scientific image processing; US National institute of Health; USA, 2022).

**Figure 9 nanomaterials-12-02735-f009:**
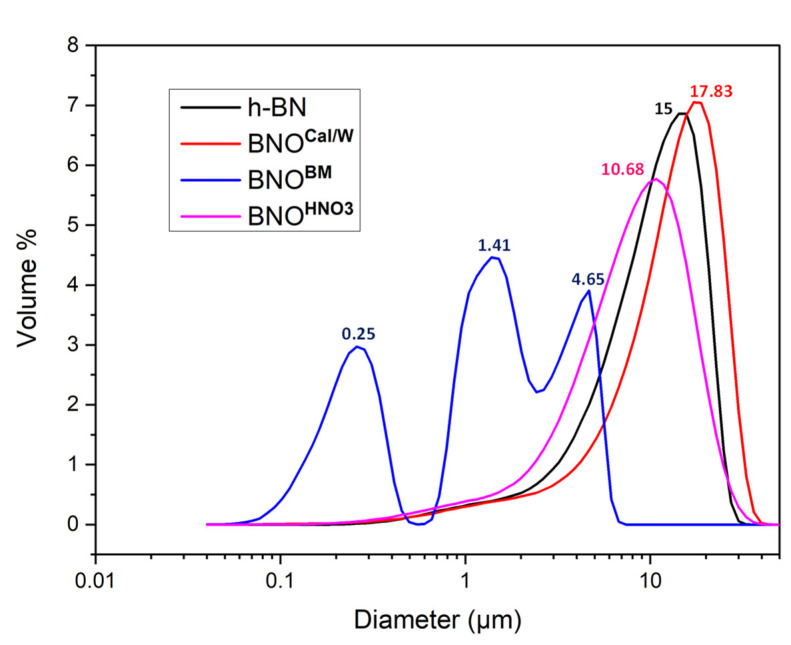
h-BN and hydroxylated BNO granulometry in water (volume %).

**Figure 10 nanomaterials-12-02735-f010:**
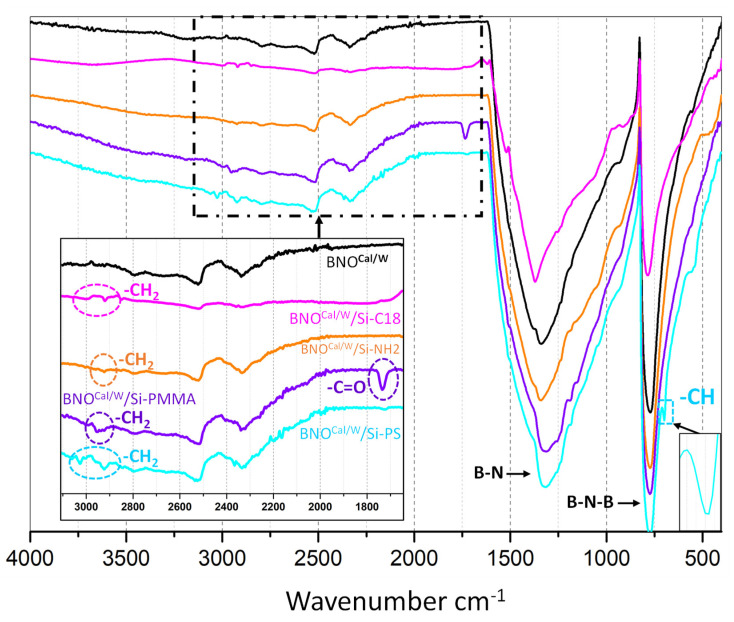
FTIR of BNO^Cal/W^ and grafted BNO^Cal/W^ with several silane grafting agents.

**Figure 11 nanomaterials-12-02735-f011:**
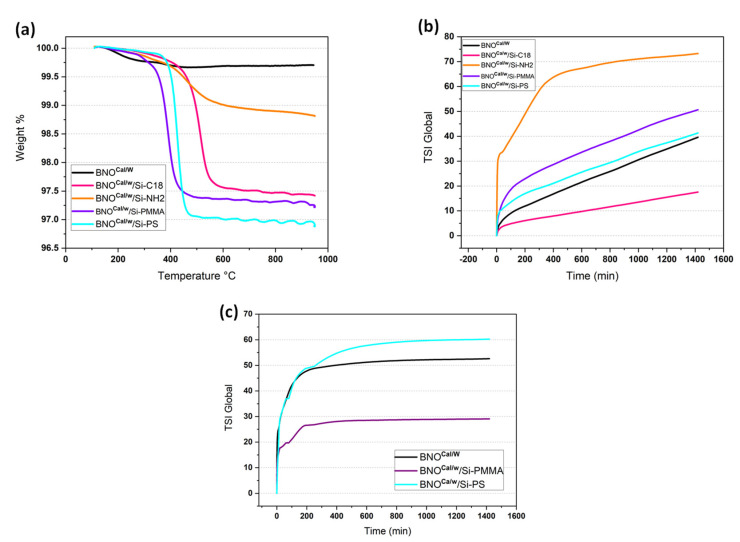
(**a**) TGA of grafted BNO^Cal/W^ compared with that of BNO^Cal/W^. (**b**) TSI global in water and (**c**) TSI global in toluene for BNO^Cal/W^ and grafted BNO^Cal/W^.

**Figure 12 nanomaterials-12-02735-f012:**
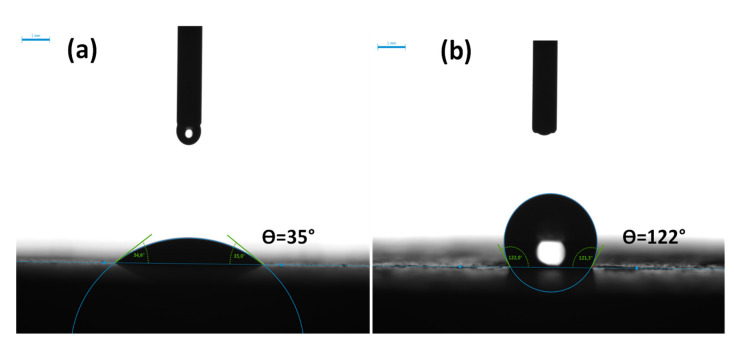
Water drop shape at the surface of (**a**) h-BN and (**b**) BNO^Cal/W^/Si-PMMA.

**Figure 13 nanomaterials-12-02735-f013:**
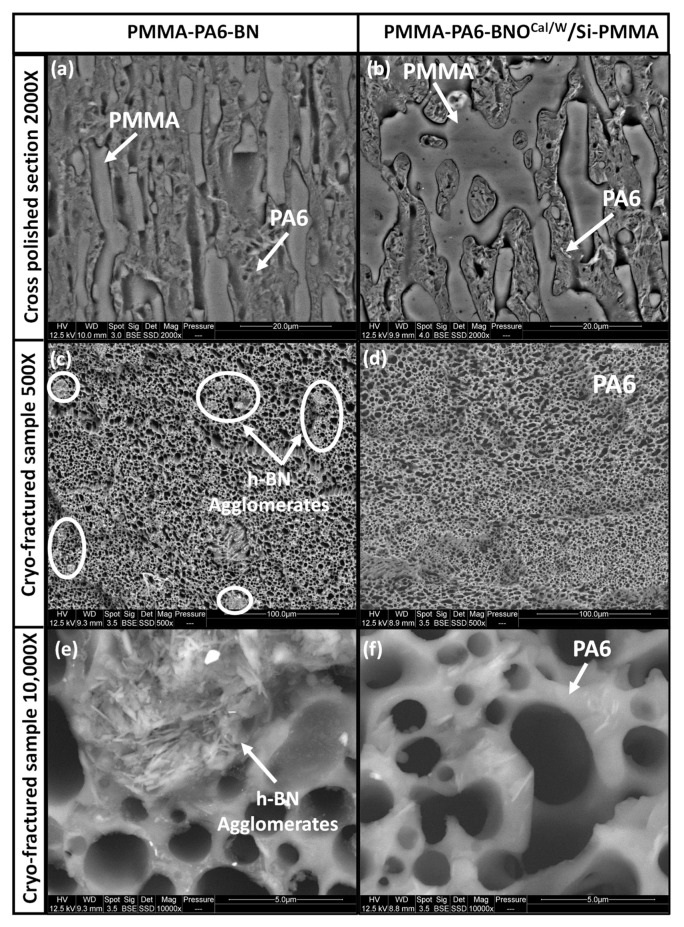
SEM images of PMMA-PA6-h-BN 8 wt% (left column) and PMMA-PA6-BNO^Cal/W^/Si-PMMA 8 wt% (right column). Parallel cross-polished sections (**a**,**b**) and perpendicular cryo-fractured surfaces after PMMA solvent extraction (**c**–**f**).

**Figure 14 nanomaterials-12-02735-f014:**
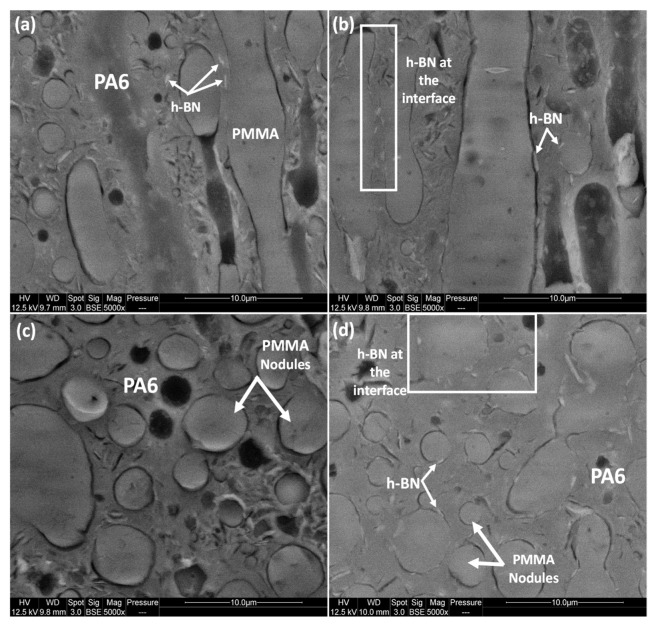
SEM images of (PMMA+h-BN)-PA6 with 8 wt% of h-BN (left column) and (PMMA + BNO^Cal/W^/Si-PMMA)-PA6 with 8 wt% of BNO^Cal/W^/Si-PMMA (right column). Parallel cross-polished sections (**a**,**c**) and perpendicular cryo-fractured surfaces (**b**,**d**).

**Figure 15 nanomaterials-12-02735-f015:**
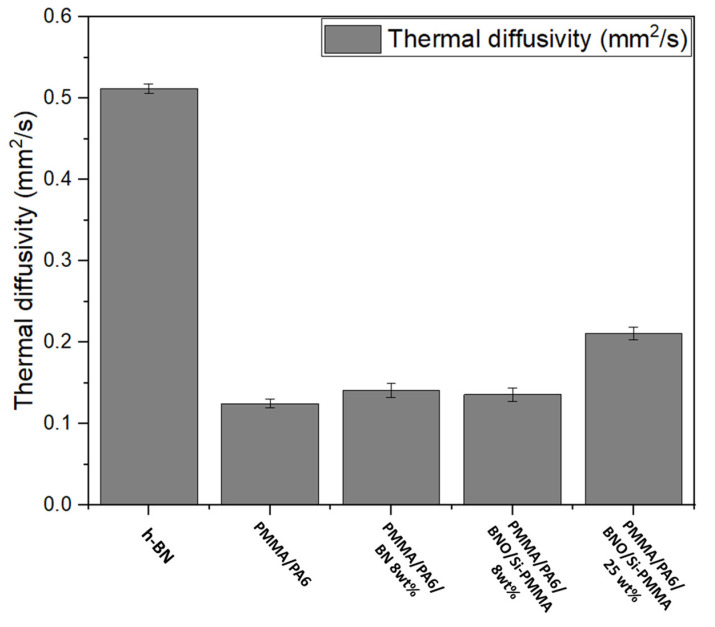
Thermal diffusivity of h-BN, PMMA/PA6, and PMMA/PA6/h-BN composites with different loading fillers.

**Table 1 nanomaterials-12-02735-t001:** List of abbreviations for the different products used in this study.

Name	Sample
h-BN	Hexagonal boron nitride
BNO	Hydroxylated boron nitride
BNO^Cal^	Hydroxylated boron nitride treated with calcination
BNO^Cal/W^	Hydroxylated boron nitride treated with calcination and washed with water
BNO^HNO3^	Hydroxylated boron nitride treated with nitric acid and sonication
BNO^BM^	Hydroxylated boron nitride treated with ball milling and using gallic acid
Si-C18	Trimethoxyoctadecylsilane
Si-NH_2_	3-aminopropyltriethoxysilane
Si-PMMA	poly(methylemethacrylate-co-3-(triethoxysilyl)propyl methacrylate)
Si-PS	poly(styrene-co-3-(triethoxysilyl)propyl methacrylate)
BNO^Cal^/Si-C18 (or BNO-C18)	Calcinated boron nitride grafted with trimethoxyoctadecylsilane
BNO^Cal^/Si-NH_2_ (or BNO-NH_2_)	Calcinated boron nitride grafted with 3-aminopropyletriethoxysilane
BNO^Cal^/Si-PS (or BNO-PS)	Calcinated boron nitride grafted with poly(styrene-co-3-(triethoxysilyl)propyl methacrylate)
BNO^Cal^/Si-PMMA (or BNO-PMMA)	Calcinated boron nitride grafted with poly(methylemethacrylate-co-3-(triethoxysilyl)propyl methacrylate)

**Table 2 nanomaterials-12-02735-t002:** Localization of particle (S) depending on wetting parameter.

Wetting Parameter Value	Localization of Particle (S)
ω_AB_ < −1	Polymer B
−1< ω_AB_ < 1	Interface
ω_AB_ > 1	Polymer A

**Table 3 nanomaterials-12-02735-t003:** Surface tension values of components of the blends.

Material	$\gamma_{i}(\mathrm{mN}/m)$	$\gamma_{i}^{d}(\mathrm{mN}/m)$	$\gamma_{i}^{p}(\mathrm{mN}/m)$
h-BN	70.43	48.17	22.26
BNO^Cal/W^/Si-PMMA	49.53	45.15	4.38
PMMA	44.38	39.46	4.92
PA6	43.44	35.42	8.03

**Table 4 nanomaterials-12-02735-t004:** Interfacial tensions and wetting parameter.

Material	γ_BN-PMMA_ (mN/m)	γ_BN-PA6_ (mN/m)	γ_PMMA-PA6_ (mN/m)	γ_BNO_^Cal/W^_/Si-PMMA-PMMA_ (mN/m)	γ_BNO_^Cal/W^_/Si-PMMA-PA6_ (mN/m)	Ꞷ_AB_ before Grafting	Ꞷ_AB_ after Grafting
Harmonic mean equation	-	-	0.96	-	-	−2.25	0.93
Geometric mean equation	6.71	4.55	-	0.23	1.13		
Prediction	-	-	-	-	-	PA6	Interface

**Table 5 nanomaterials-12-02735-t005:** Samples composition.

Sample Name	Filler Type	Filler (wt%)	PMMA (wt%)	PA6 (wt%)
PMMA-PA6-h-BN	h-BN	8	46	46
PMMA-PA6-BNO^Cal/W^/Si-PMMA	BNO^Cal/W^/Si-PMMA	8	46	46
(PMMA/BN)-PA6	h-BN	8	46	46
(PMMA/BNO^Cal/W^/Si-PMMA)-PA6	BNO^Cal/W^/Si-PMMA	8	46	46

## Data Availability

All data are reported in this article.

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
