# Peer review of "h-BN Modification Using Several Hydroxylation and Grafting Methods and Their Incorporation into a PMMA/PA6 Polymer Blend"

_nanomaterials, 2022, doi:10.3390/nano12162735_

Round 1
Reviewer 1 Report
The manuscript reported on the modification of hBN particles by hydroxylation and grafting methods to improve their dispersion into a PMMA/PA6 polymer blend. The research is an important step toward the development of hBN/polymer composites with a high thermal conductivity. The authors carried out several hydroxylation and grafting methods and thorough characterization of the products to demonstrate the successful hydroxylation and grafting. An obvious improvement of hBN dispersion in the PMMA/PA6 polymer blend is achieved. Therefore, I recommend the acceptance of the manuscript after the authors address the following concerns:
1. Many figures in the manuscript are missing proper labels. For example, the corresponding functional groups for important peaks in all the FTIR spectra should be clearly labeled. In the hBN/PMMA/PA6 composites, clear labels in many SEM images are missing, but they should be provide to demonstrate the distribution of hBN, PMMA, and PA6.
2. In Figure 4b, there are 2 obvious weight loss stages for BNO(HNO3). Comments should be provided.
3. The authors carefully investigated the stability of the modified hBN particles in water. However, water is usually not used when incorporating hBN particles into polymers. Therefore, why this kind of research is important should be clearly stated.
4. There is an obvious difference in the particle size distribution in Figure 8 and 9. Comments should be provided.
5. Explanations and labels for Figure 10c should be provided.
Author Response
you can find the response point by point to your comments and the one of the other reviewer in the attached file

Reviewer 2 Report
Boukheit and co-workers used nitric acid-assisted ball milling for the surface functionalization of boron nitride (BN), which subsequently react with silane-grafted copolymers and determined the material properties of resulting BN-polymer composites. Although in the beginning of the highlights the author claimed “this strategy has not been explored to obtain thermally conductive composites”, many groups have already published articles on similar topics and methods (for example, a review article previously published in Chem. Soc. Rev., 2016, 45, 3989-4012). So, the paper should describe the improvement of thermal conductivity and mechanical properties in BN-polymer composites which I did not observe. In addition, a research strategy based on the reaction between the hydroxyl-functionalized BN and silane-grafted copolymers is similar to previously published work by the authors (Polymer, 2020, 191, 122277). However, after carefully reading the manuscript from the beginning to the end, I think this work is more appropriate for more specialized MDPI journals in the fields of polymer science and composites.
Author Response
you can find attached the response point by point to your comments and the one of the other reviewer

Round 2
Reviewer 2 Report
The revised manuscript has slightly changed with providing some explanations, however I mentioned some suggestions in my previous review report, but the authors failed to improve on these suggestions. Thus, I am still very convinced that this manuscript cannot be accepted for publication in its current form.
Major comments:
1. After carefully reading the revised manuscript, I can fully understand the authors' argument and purpose. However, thermal conductivity and mechanical properties of these composites are unknown to reader in this study. So, theses results should be provided in the main text.
2. Merit of the manufacturing process and physical performance of the resulting composites should be compared those from with recently published papers.
Author Response
Reviewer 2 mentioned in his second report that we failed to take into account some of his suggestions. Then he developed some two major changes that have to be taken into account:
- He suggests to provide some mechanical and thermal properties
- He suggests to add recently published articles on the manufacturing process and physical performances of similar composites
To answer point 1, we understand that it is important to add some final properties. Mechanical properties are not important for the intended applications. About thermal conductivity properties, thermal diffusivity was measured on several samples. As the results have to be consolidated we decided to place those results in the supporting information. Please see the highlighted part (S6, Table S1).
To answer to point 2, the reviewer knows that it is not possible to give a complete review of all articles dealing with the objectives of the present articles. Our article contains 72 references (with very recent references), which is a high number for a research article . Moreover, I want to insist that the objectives of the present article are not to fabricate thermally conductive materials but to compare different ways to hydroxylate and functionalize h-BN and to see the influence on the final polymer blend morphology. On this topic, even if we were not exhaustive, many references are cited .
Finally, we hope that the reviewer 2 will understand our arguments regarding its comments.
